# Ionomer-free and recyclable porous-transport electrode for high-performing proton-exchange-membrane water electrolysis

Jason K. Lee[1], Grace Anderson[1,2], Andrew W. Tricker [1], Finn Babbe [1],
Arya Madan[2], David A. Cullen [3], José' D. Arregui-Mena[3], Nemanja Danilovic[1],
Rangachary Mukundan[1], Adam Z. Weber [1] & Xiong Peng [1] ✉

Clean hydrogen production requires large-scale deployment of water-electrolysis technologies, particularly proton-exchange-membrane water electrolyzers (PEMWEs). However, as iridium-based electrocatalysts remain the only practical option for PEMWEs, their low abundance will become a bottleneck for a sustainable hydrogen economy. Herein, we propose high-performing and durable ionomer-free porous transport electrodes (PTEs) with facile recycling features enabling Ir thrifting and reclamation. The ionomer-free porous transport electrodes offer a practical pathway to investigate the role of ionomer in the catalyst layer and, from microelectrode measurements, point to an ionomer poisoning effect for the oxygen evolution reaction. The ionomer-free porous transport electrodes demonstrate a voltage reduction of > 600 mV compared to conventional ionomer-coated porous transport electrodes at 1.8 A cm$^{-2}$ and <0.1 mg$_{Ir}$ cm$^{-2}$, and a voltage degradation of 29 mV at average rate of 0.58 mV per 1000-cycles after 50k cycles of accelerated-stress tests at 4 A cm$^{-2}$. Moreover, the ionomer-free feature enables facile recycling of multiple components of PEMWEs, which is critical to a circular clean hydrogen economy.

Clean hydrogen, if produced by water electrolysis using renewable electricity, stands at the epicenter of decarbonizing various hard-to-decarbonize sectors that rely heavily on fossil fuels[1], and attaining the global goal of restricting temperature rise below 1.5 °C[2]. In the transportation sector, hydrogen fuel cells are promising alternative to traditional diesel combustion engines for deployment of heavy-duty vehicles because of the simplicity in scaling power and energy[3,4]. Hydrogen also plays a critical role in decarbonizing industrial sectors, which currently account for 12% of global emissions[5]. The iron and steel as well as ammonia productions are one of the largest contributors for

industrial $CO_2$ emissions, accounting for approximately 30% and 11% of global industrial emissions, respectively;[6,7] and recent studies show that hydrogen can provide solutions in curtailing these industrial $CO_2$ emissions[6–9].

The most promising technology for production of clean hydrogen is the proton-exchange-membrane water electrolyzer (PEMWE) as it provides many advantages such as production of high purity hydrogen (99.999%) at high pressure (up to 30 bar), compact system design, and wide operating window[10,11]. Utmost importance of PEMWE is that it provides robust dynamic response to the applied load, offering great

[1]Energy Storage and Distributed Resources Division, Lawrence Berkeley National Laboratory, Berkeley, CA 94720, USA. [2]Department of Chemical and Biomolecular Engineering, University of California Berkeley, Berkeley, CA 94720, USA. [3]Center for Nanophase Materials Sciences, Oak Ridge National Laboratory, Oak Ridge, TN, USA. ✉e-mail: xiongp@lbl.gov

synergy with highly intermittent renewable energy sources compared to conventional liquid-alkaline water electrolyzers[12,13]. Technical maturity of PEMWEs has been significantly improved over the years, with successful deployment of commercial stacks at kilowatt (kW) to megawatt (MW) scales[14]. However, worldwide demand for hydrogen is projected to increase from 90 million tonnes per annum (Mtpa) in 2020 to over 500 Mtpa by 2050, which requires electrolyzer deployments at gigawatt (GW) to terawatt (TW) levels[15].

Industry encounters multiple barriers in scaling up from MW to GW scale. Even with the ambitious assumption that electricity powering GW scale plants comes from inexpensive renewable resources, there are still significant capital expenditures to realize GW-scale electrolysis. For instance, the amount of titanium used for bipolar plates and anode porous-transport layers (PTLs) will be directly scaled with the increase in active area or number of cells in a stack. The throughput required in processing catalyst layers—the processes of catalyst synthesis, ink fabrication, and catalyst-layer coating—must keep pace with the scale-up. The largest bottleneck is likely to come from the limited natural abundance of platinum-group metals (PGMs). Due to the highly corrosive and acidic operating environment on the anode side that performs the oxygen-evolution reaction (OER), iridium (Ir) or Ir-based catalysts are the only feasible anode catalyst for commercial PEMWEs for high durability and activity[10]. However, Ir is the rarest element on earth, which is mined only 7 tons per annum[16]. Even assuming half of the annual global Ir supply is dedicated to electrolysis application, only a few electrolyzer plants at a GW scale would be installed, meaning that it will not be able to supply the increased demand of carbon-free hydrogen. Therefore, developing Ir electrodes at low loading (0.1 ~ 0.4 $mg_{Ir}$ $cm^{-2}$) with facile recyclable features for high-performing PEMWEs is critical for successful deployment of GW-scale systems and creating circular clean hydrogen economy.

Recent studies have made significant progress in reducing the amount of anode catalyst loading used in electrolyzers. From the traditional high loadings of 1 ~ 3 $mg_{Ir}$ $cm^{-2}$, stable performance has been demonstrated at ultra-low loadings of 0.08 $mg_{Ir}$ $cm^{-2}$ and below[17–20]. For example, nano-structured thin film catalysts manufactured by 3 M[21], have demonstrated excellent performance under low loading (<0.25 $mg_{Ir}$ $cm^{-2}$), however novel electrodes have been relatively underexplored compared to conventional catalyst layers consisting of ionomers and nanoparticles. While advancements have been made

towards loading reduction using direct electrode deposition techniques such as works by Yasutake et al.[22], and Hrbek et al.[23], MEA performance is not yet comparable to the conventional catalyst layers, and recyclability of these anode catalysts has been under investigated. Traditional processes implemented for PGM extraction and recovery, such as pyrometallurgical and hydrometallurgical are not well suitable for recycling catalyst materials because of fluorine-contained ionomer used in conventional catalyst layers, which can co-produce hazardous emissions and might need to be recycled itself[24]. Recycling platinum nanoparticles via electrochemical dissolution-electrodeposition has been recently demonstrated by Sharma et al.[25], which shows promising results towards recycling infrastructure in the near future.

In this work, we propose promising solutions to the challenge of limited Ir availability and catalyst recycling with our ultra-low loaded ionomer-free porous-transport electrode (PTE). The proposed PTE and its fabrication eliminate the need for ionomer in the anode compartment, thereby enabling easier recycling, while also demonstrating excellent activity and durability at low loading (<0.1 $mg_{Ir}$ $cm^{-2}$) for PEMWEs.

## Results

### Fabrication and structure of ionomer-free PTEs

Fabrication of traditional catalyst-coated membranes (CCMs) or PTEs requires at least five steps: mining, refining, synthesizing, fabricating, and coating (Fig. 1). First, Ir must be mined from the Earth's crust, which is available only in a few geographical regions including South Africa, Russia, United States, Canada, and Zimbabwe[16]. Once the Ir ore has been gathered, it must undergo complicated procedures of refining and processing, such as removal of crude ores and separation of PGM concentrate from other metals and rocks. Then, these concentrates are further separated to acquire high purity Ir metal suitable for catalytic applications[26]. At this stage, high-purity Ir metals are further processed through series of transformations to achieve the precursor form, which can be synthesized into nanoparticles to be used as water-splitting catalysts, after which, electrolyzer manufacturers mix iridium catalyst, ionomer, and solvents to formulate catalyst inks followed by ink coating to fabricate PTEs or CCMs. Although an ultrasonic-based coating method is known to produce a more uniform and high-quality Ir catalyst layer, even at ultra-low loadings, it is a low throughput and economically unfavorable method due to the use of

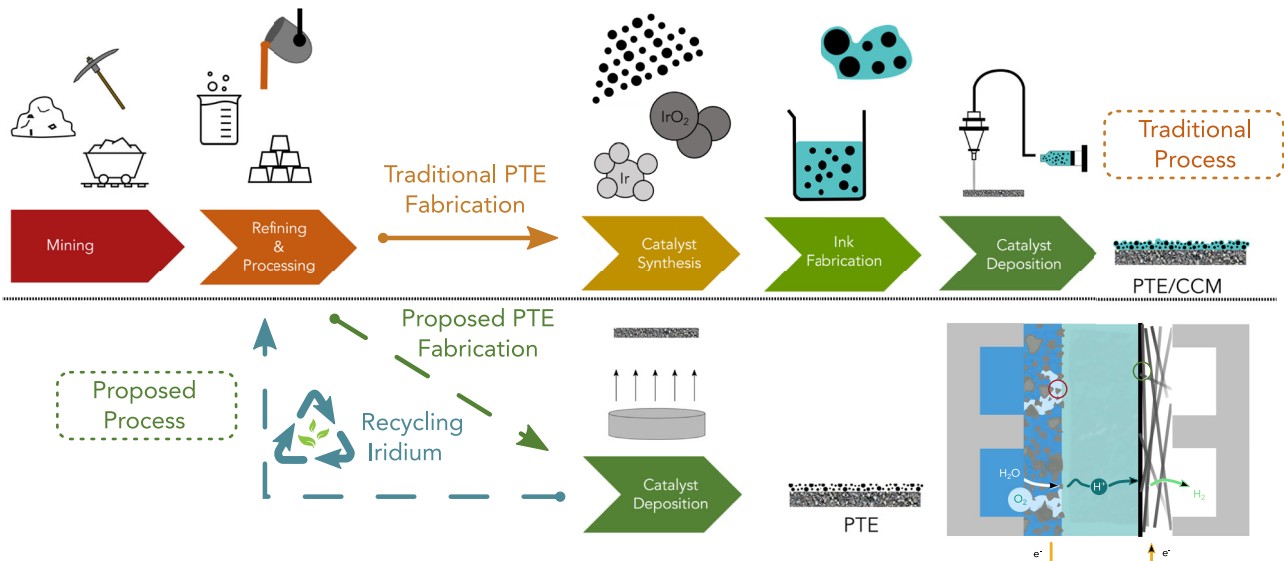

**Fig. 1 | Comparison between conventional ways of CCM/PTE fabrication process and the ionomer-free PTE fabrication.** The upper flow-chart describes the traditional process of PTE/CCM fabrication, and the lower flow-chart describes the proposed ionomer-free PTE fabrication. A schematic of the PEM electrolyzer is shown in the right bottom corner.

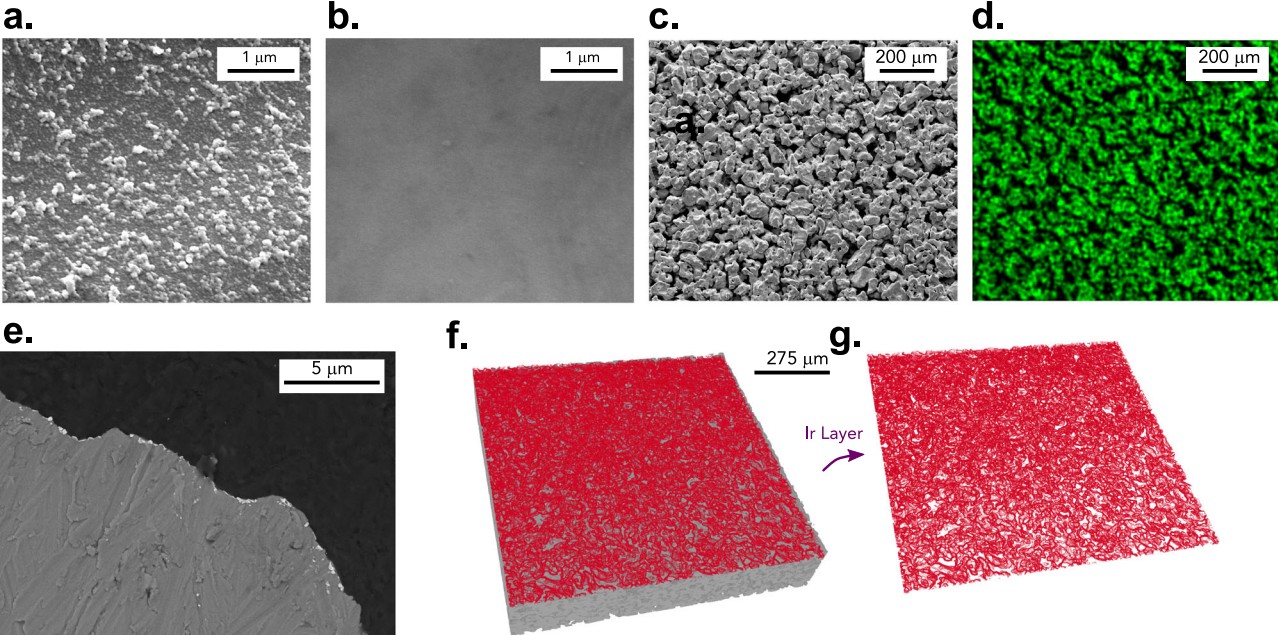

**Fig. 2 | Structural and morphological features of the ionomer-free PTE. a** SEM image of the Ir coated PTE, showing the presence of Ir nanoparticles. **b** Uncoated PTL. **c** Zoomed-out SEM image of the PTE. **d** Correspondent EDS image of **c**, showing the distribution of Ir across PTE surface. **e** Focus-ion beam scanning electron microscope (FIB-SEM) image showing the cross section of the ionomer-free Ir PTE. **f** Synchrotron XCT image of the PTE. **g** Coated Ir-layer obtained from **f**.

very dilute inks[27]. In contrast, rod coating[28] and roll-to-roll[29] methods can manufacture electrodes at high throughput, yet they are limited to usage of viscous inks with high solid contents for fabricating electrodes at high loadings.

Previous studies indicate that anode catalyst layer in-plane electronic conductivity limits the catalyst utilization especially at low loading, therefore necessitating improved interfacial PTL/catalyst-layer contact[30–32]. Strategies such as adding microporous layer (MPL) or direct interfacial modification have been applied to improve PEMWEs performance[33,34]. Compared to CCMs, PTEs can have advantages in forming a better contact between catalyst layer and PTL by reducing catalyst layer deformation upon membrane swelling, therefore allowing better catalyst utilization especially at low loading. However, direct ink coating on the PTL to fabricate PTE often encounters challenges due to the porous nature of the PTL, leading to ink intrusion and low PEMWE performance. Our proposed PTE fabrication technique eliminates two processes from the traditional fabrication technique: i) catalyst synthesis and ii) catalyst-ink fabrication. After refining of the iridium, it is made into a form of a target, which we use to directly coat a layer of iridium to the PTL by physical vapor deposition (PVD), forming a nanosized catalyst layer well adhered to the PTL, eliminating the use of ionomer binder to maintain catalyst-layer integrity, which reduces capital cost. The direct Ir PVD method is scalable as it is facile and is a mature process for industrial uses such as in anti-reflective[35], textile[36], and solar-cell applications[37], and because it is a line-of-sight method, it only coats catalyst at the interface instead of coating the entire PTL inner and outer surfaces as it would do in other processes (e.g., electroplating).

The surface morphology of the ionomer-free Ir PTE is shown in Fig. 2. Microscopically, a thin layer of Ir particles is coated on top of the PTL surface (Fig. 2a vs. 2b, PTE vs. PTL). Zoomed-out scanning electron microscope (SEM) image (Fig. 2c) of PTE indicates that the surface macroscopic features of PTL remains unchanged after PVD coating. The correspondent Energy-dispersive spectroscopy (EDS) measurement (Fig. 2d) confirms a uniform Ir coating only at the surface of the PTL, while showing no presence of Ir at open surface pores; therefore, indicating negligible Ir penetrating into the PTL during coating. A small

oxide peak in the Ir phase is shown by X-ray photoelectron spectroscopy (XPS) measurement, however the X-ray diffraction (XRD) measurement only reveals Ir metal diffraction peaks – indicating that the Ir oxide layer formed on the PTE is either too thin to be detected through XRD or amorphous (Supplementary Fig. 1). The cross-sectional image from FIB-SEM verifies that a thin Ir layer is achieved after coating at typical thickness of 90–100 nm (Supplementary Fig. 2), and it indicates the uniformity of the coated catalyst layer at such a low loading of 0.085 $mg_{Ir}$ $cm^{-2}$ (Fig. 2e), as does the synchrotron X-ray computed tomography (XCT) images of the PTE (Fig. 2f, g). The impact of thermal annealing at various temperatures on ionomer-free PTE is also investigated, the result indicates that extra posttreatment is not needed, which simplifies the ionomer-free PTE fabrication (Supplementary Discussion 1, Supplementary Figs. 3 and 4).

**PEMWE performance and the role of ionomer**

We first compare the ionomer-free PTE with that of a conventional PTE at the same anode loading conditions of ~0.1 $mg_{Ir}$ $cm^{-2}$, fabricated by coating catalyst ink directly to PTL surface (defined as sprayed-PTE) using an ultrasonic sprayer. As shown in Fig. 3a, the sprayed-PTE performed significantly worse compared to the ionomer-free PTE, with much higher ohmic and mass-transport overpotentials compared to the ionomer-free Ir PTE (Supplementary Fig. 5). Moreover, the ionomer-free Ir PTE provides substantially better electrode kinetics, as demonstrated by lower kinetic overpotential (Fig. 3b) and lower Tafel slope (Fig. 3c). Direct ink coating to PTL surface leads to catalyst-ink penetration through surface pores of the PTL and depositing within the PTL—thereby losing active catalysts and resulting in inhomogeneous coating as well as extremely low catalyst utilization (Supplementary Fig. 6). In contrast, the ionomer-free PTE has Ir coating only at the interfacial surface of the PTL, leading to negligible catalyst loss and thus maximizing deposited catalyst utilization. The method of direct Ir coating not only curtails the cost of PTE manufacturing processes but also eliminates safety hazards because it does not require catalyst-ink fabrication, which use highly flammable solvents at industrial scales[38].

To investigate the impact of ionomer in PTEs on PEMWE performance, we coat an additional layer of perfluorosulfonic-acid (PFSA)

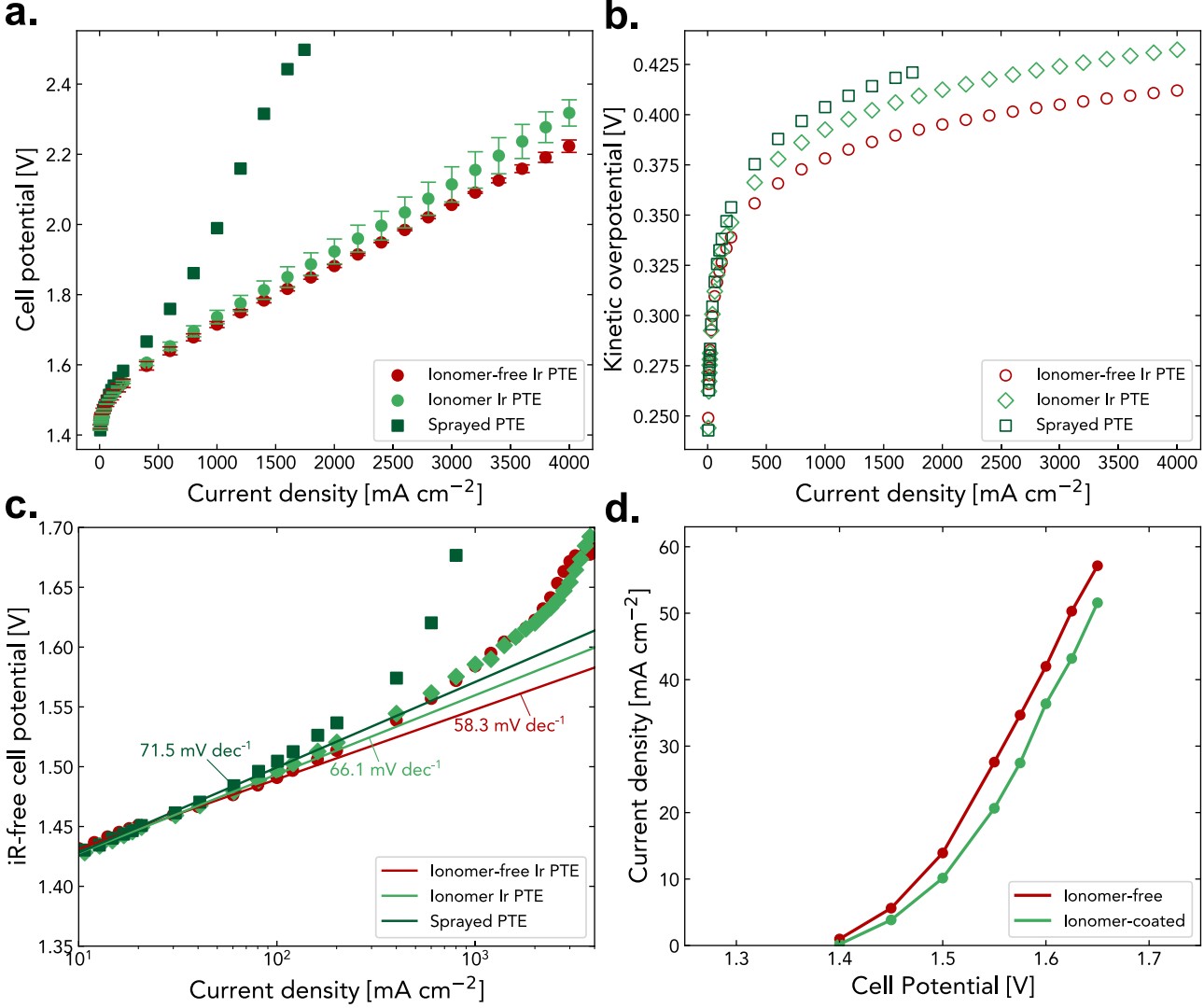

**Fig. 3 | Electrochemical performance of various PTEs.** Comparison of **a** polarization curves and **b** kinetic overpotentials. **c** Measured Tafel slopes among the ionomer-free PTE, ionomer-coated PTE, and traditional ultrasonic spray coated PTE. **d** Polarization curves of the ionomer-free and ionomer-coated Ir for oxygen-evolution reaction in a microelectrode setup. The error bars represent the spread between two independent experiments.

ionomer (Nafion) on the Ir PTE surface (defined as ionomer Ir PTE) at the same anode loading conditions (Supplementary Fig. 6). Surprisingly, the ionomer Ir PTE exhibited higher overpotentials compared to the ionomer-free Ir PTE (Fig. 3a), with elevated ohmic loss (Supplementary Fig. 5) especially at high current densities. This is likely due to that higher oxygen-transport resistance through the ionomer phase leads to oxygen bubble accumulation near catalyst surface at high currents, which in return can impact reactant water supply and results in local dehydration of the polymer electrolyte. Besides, the ionomer-coated Ir PTE exhibits more sluggish electrode kinetics compared to ionomer-free PTE, as indicated by higher kinetic overpotential (Fig. 3b) and Tafel slope (Fig. 3c). The higher kinetic overpotential is suspected to be due to potential ionomer poisoning of the catalyst surface. In the case of PEM fuel cells, the specific adsorption of ionomer sulfonic-acid groups on Pt surface has also been suggested to poison catalyst active sites, impacting the oxygen-reduction-reaction kinetics at least from rotating disk electrode (RDE) tests[39–41]. We therefore hypothesize that similar ionomer adsorption behavior can occur to Ir surface, which could impair OER kinetics. A microelectrode setup (Supplementary Fig. 7) is used to further demonstrate the ionomer poisoning effect to Ir[42,43]. Microelectrode has a very small active area (75 μm diameter) and thus has a very low total current (Supplementary Fig. 8), minimizing

the contribution from ohmic losses. Therefore, the difference in the performance is dominated by electrode kinetics and mass transport. Compared to conventional RDE measurement, microelectrode primarily relies on the catalyst/polymer-electrolyte interface for charge transfer, which is more representative to a membrane-electrode assembly (MEA) devices. A bare Ir microelectrode (ionomer-free) was first utilized to measure the OER activity followed by re-measuring after dipping in Nafion ionomer dispersion to make the ionomer-coated microelectrode. The OER polarization curve obtained using microelectrode indicates an inhibited OER activity for ionomer-coated Ir compared to the ionomer-free Ir (Fig. 3d). Both the MEA and microelectrode results indicate that ionomer in the catalyst layer can impact electrode kinetics, potentially through a poisoning effect for oxygen evolution reaction.

**Impact of Ir loadings and interfaces on PEMWE performance**

Iridium loading reduction is a crucial step towards realizing GW-scale electrolyzers. Our proposed ionomer-free Ir PTE is well suited for low loading and ultra-low loading conditions. The electrochemical performance of the ionomer-free Ir PTEs under four different Ir loadings of 0.033, 0.050, 0.085, and 0.187 $mg_{Ir}$ $cm^{-2}$ is given in Fig. 4. The applied-voltage breakdown of ultra-low loaded ionomer-free Ir PTEs is

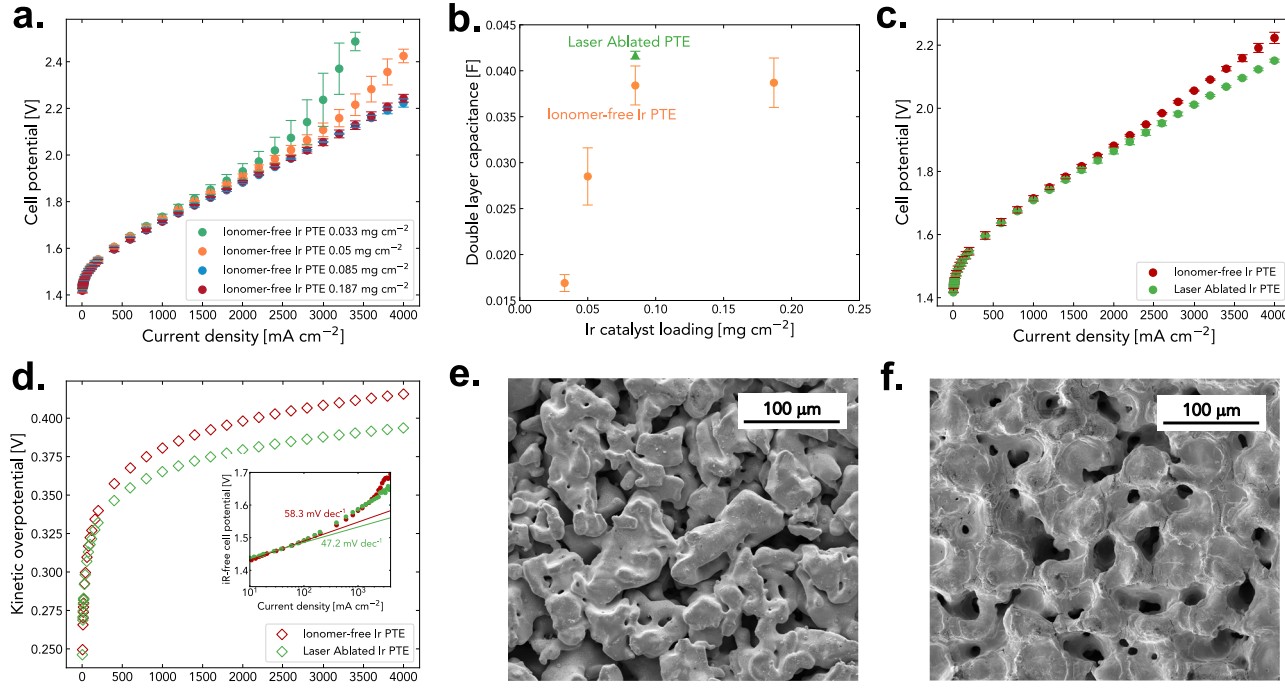

**Fig. 4 | The impact of catalyst loading on the ionomer-free Ir PTE and the improved electrolyzer performance with enhanced interface via laser ablation.** **a** Polarization curves for ionomer-free Ir PTE at 0.033, 0.050, 0.085, and 0.187 mg$_{Ir}$ cm$^{-2}$. **b** double-layer capacitance measured at different loadings and the laser ablated ionomer-free Ir PTE. **c** Polarization curve. **d** Kinetic overpotential measured for the laser ablated ionomer-free Ir PTE at 0.085 mg$_{Ir}$ cm$^{-2}$. The inset image plots Tafel slopes for the laser ablated and baseline ionomer-free Ir PTE (47.2 mV dec$^{-1}$ vs 58.3 mV dec$^{-1}$). **e** SEM image of bare PTL surface. **f** SEM image of laser ablated PTL surface. The error bars represent the spread between two independent experiments.

found in Supplementary Fig. 9. As the Ir loading increases from 0.033 to 0.085 mg$_{Ir}$ cm$^{-2}$, there is an enhancement in electrolyzer performance; however, further increase in loading to 0.187 mg$_{Ir}$ cm$^{-2}$ leads to negligible performance improvement, which is likely due to a formation of a smooth and dense catalyst layer on the PTE (Supplementary Fig. 10), reducing the surface area of the Ir layer which dwindles the number of active sites available for the OER. The increase in double-layer capacitance plateaus after 0.085 mg$_{Ir}$ cm$^{-2}$ (Fig. 4b), suggesting that increasing Ir loading is not the most efficient way to improve the ionomer-free Ir PTE performance. Instead, we explore other possible ways through engineering the PTL surface.

As the Ir layer is supported by the PTL to form PTE, one pathway to maximize the number of active sites is to improve the surface roughness of PTL, where more Ir can adhere to. Here we utilize a laser ablation technique to achieve a rougher PTL surface. Specifically, the molten structure of the titanium created by the heat from the laser ablation closes smaller pores existing at the PTL surface, resulting in an increased surface area for electrochemical reaction to occur (Fig. 4e vs. 4f, bare PTL vs. laser ablated PTL). Compared to the baseline ionomer-free Ir PTE, the PTE fabricated using laser ablated PTL (defined as laser ablated ionomer-free Ir PTE) exhibits improved performance (maximum voltage reduction of 56 mV at 4 A cm$^{-2}$) throughout the current densities tested in the polarization curve (Fig. 4c). This is mostly due to enhanced electrochemical surface area of the Ir layer (Fig. 4b), leading to lower kinetic overpotential (Fig. 4d) and Tafel slope (47.2 mV dec$^{-1}$ compared to 58.3 mV dec$^{-1}$, inset of Fig. 4d). To further demonstrate the effectiveness of laser ablation to improve ionomer-free Ir PTE performance, we ablated fiber based PTL as shown in Supplementary Fig. 11. The ionomer-free Ir PTE fabricated using laser ablated fiber PTL (Supplementary Fig. 11b) shows improved electrolysis performance at maximum voltage reduction of 53 mV at 4 A cm$^{-2}$ (Supplementary Fig. 11c) compared to the baseline fiber PTL (Supplementary Fig. 11a), which is largely driven by improved electrode kinetics (Supplementary

Fig. 11d) through improved interfacial area. The performance of ionomer-free Ir PTE under ultra-low loading presented in this work outperforms state-of-the-art PTEs operating at similar conditions (N117, 80 °C) reported in literature with a 28-fold decrease in Ir loading (see Supplementary Table 1)[33,44–50].

## Durability of the ionomer-free Ir PTE

As long-term stability is an essential piece of the electrode design, the stability of ionomer-free PTE is evaluated through an accelerated stress test (AST). A square wave potential cycling between 1.45–2.2 V was applied to the ionomer-free Ir PTE with 5 s hold at each potential. The AST protocol was selected based on a comprehensive study by Alia et al.[51], where they found that it induced more catalyst-layer degradation compared to constant current hold. Hence, this AST protocol is an indicator of the catalyst durability of the ionomer-free Ir PTE electrolyzer. A 5 cm$^2$ ionomer-free Ir PTE was assembled in a PEMWE and underwent a total of 50k AST cycles with polarization curve and galvanostatic electrochemical impedance spectroscopy (EIS) measurements recorded in between.

The polarization curve acquired from the AST shows change of only 29 mV difference at 4 A cm$^{-2}$ after 50k cycles (Fig. 5a) at average rate of 0.58 mV per 1000-cycles, indicating excellent durability of the ionomer-free PTE for PEMWE application when comparing to the literature reporting degradation rate of 4.51 mV per 1000-cycles at 2 A cm$^{-2}$ in a CCM configuration with Ir loading of 0.1 mg$_{Ir}$ cm$^{-2}$ at similar conditions[51]. Even though a small gradual increase in Tafel slopes is observed during ASTs, its impact on performance has been largely offset by decrease in high frequency resistance (HFR) (Fig. 5b) perhaps through continued conditioning, leading to an overall insignificant performance penalty during AST. Measured EIS also demonstrated no sign of noticeable degradation after 50k cycles (Supplementary Fig. 12). The XRD patterns of PTE after AST exhibited negligible difference compared to pristine PTE (Fig. 5c), indicating the

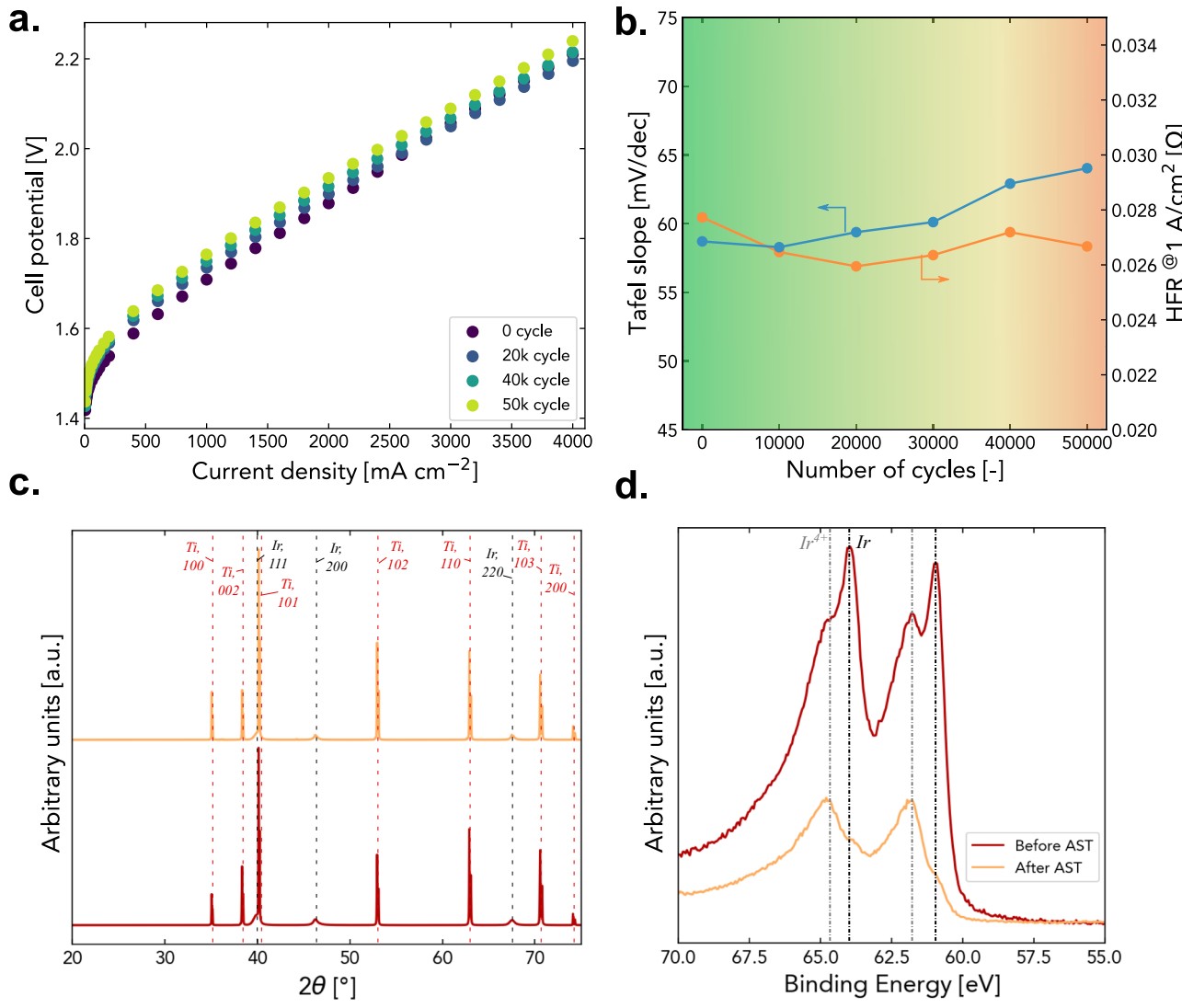

**Fig. 5 | Durability evaluation of ionomer-free Ir PTE in PEMWE using accelerated stress test up to 50k cycle. a** Polarization curves recorded at various cycles through ASTs. **b** Tafel slopes and HFR measured through AST. **c** Comparison of XRD patterns. **d** XPS measurements of Ir before and after 50k cycles of AST.

bulk phase of ionomer-free Ir PTE remained intact through the AST. XPS of the Ir after the AST shows dominant presence of Ir oxides compared to pristine PTE (Fig. 5d), while no obvious Ir oxides peak is observed through XRD, indicating a growth of amorphous surface oxides thickness, which explains the gradual increase of Tafel slopes mentioned above. X-ray fluorescence (XRF) mapping of Ir illustrates that the PTE still has uniform distribution of Ir-layer coated on the PTL surface even after 50k AST cycles (Supplementary Fig. 13). The remaining Ir loading of the PTE after AST combines the Ir loading transferred to membrane side make up to a total of 0.075 mg$_{Ir}$ cm$^{-2}$ measured using XRF, showing a total loss of just 0.01 mg$_{Ir}$ cm$^{-2}$ (vs. 0.085 mg$_{Ir}$ cm$^{-2}$) after 50k cycles. A constant-current hold durability test was also performed using ionomer-free PTE to ensure stable operation (Supplementary Fig. 14). These results clearly indicate that the ionomer-free Ir PTE exhibits outstanding durability for PEMWE applications.

### Recyclability of the Ionomer-free Ir PTEs

Recycling is a prerequisite to deployment of GW-scale electrolysis plants but has been fairly under investigated in the field. While recycling PGM materials used in catalysts are of utmost importance, recycling other costly components, such as membrane or porous

transport layers, also significantly reduces cost for large scale applications. This section studies the feasibility of recycling PTEs and half CCMs used in the PEM electrolyzer without going through excessive processes such as pyrometallurgical or hydrometallurgical extractions. Eliminating the ionomer in the catalyst layer greatly simplifies the recycling process compared to conventional MEA, as it avoids the consequences of generating toxic pollutant and need for ionomer recycling from fluorine moieties[52]. Besides, the PTE configuration naturally has benefits in recycling the PEM as it avoids catalyst coating to the membrane side during fabrication. Post AST testing, we looked into three potential recycling scenarios: i) replacing the degraded PTE with a pristine PTE and pairing with the used half CCM (Nafion 117 + cathode), ii) recoating Ir catalysts over degraded Ir catalyst layer of PTE to after AST and pairing with fresh half CCM, and iii) recoating the Ir catalysts on the other side of the PTE after AST (i.e., coating on the side originally facing flow field) and pairing with fresh half CCM. These three scenarios have similar Ir loadings of 0.085 mg$_{Ir}$ cm$^{-2}$ and investigate different pathways of recyclability. The first scenario seeks for the feasibility of expanding the lifetime of half CCM by replacing the degraded anode PTE with a fresh PTE, as degradation of MEA often comes from anode side due to harsh OER environments. Improving lifetime of half CCM also adds significantly to the cost reduction as Pt is

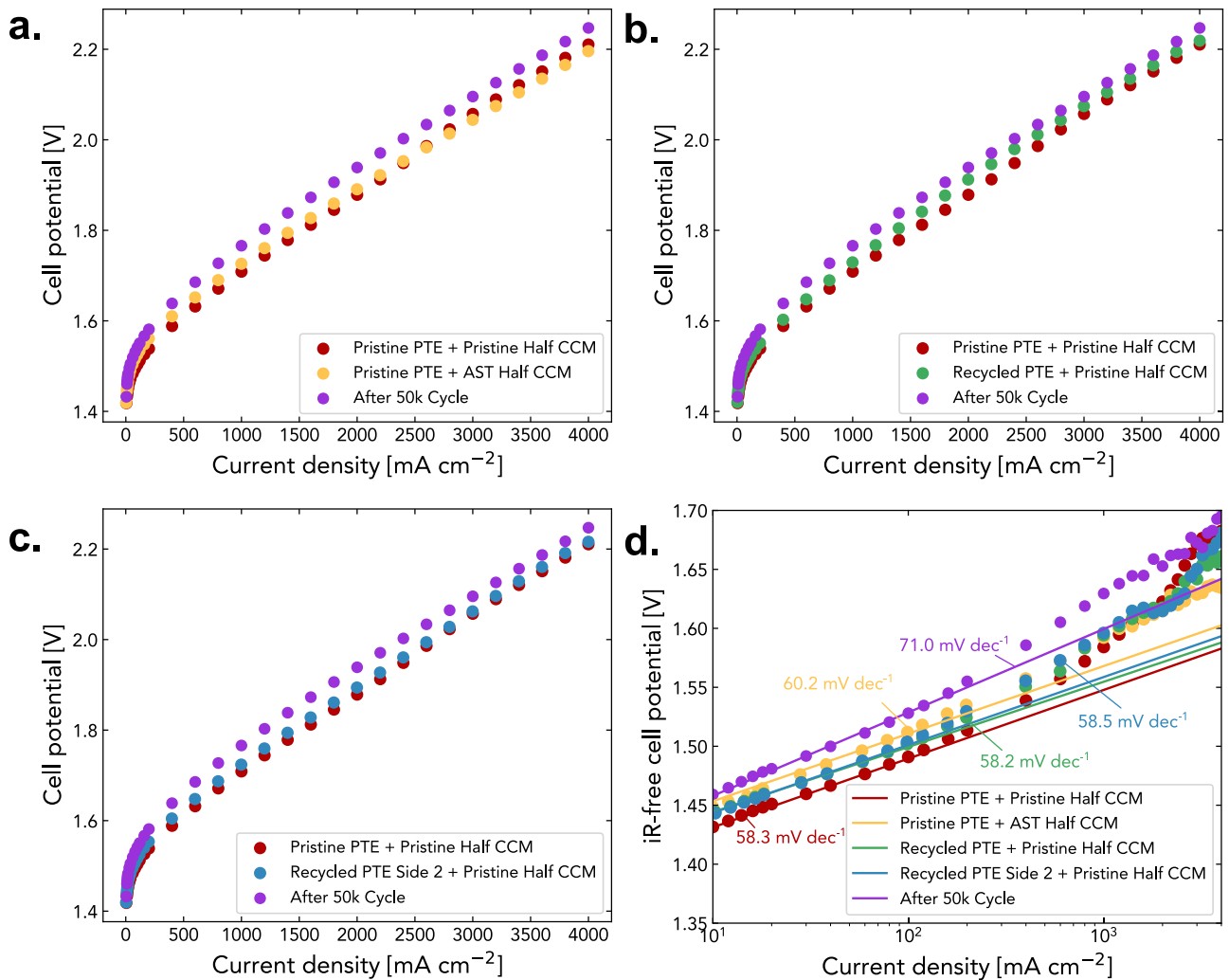

**Fig. 6 | Comparing electrolyzer performance with recycled catalysts based on different recycling scenarios. a** Replacing degraded PTE with a pristine PTE. **b** Recycling degraded PTE by reapplying Ir coating and pairing with a pristine half CCM. **c** Recycling PTL by reapplying Ir coating on the back side and using a pristine half CCM. **d** Measured Tafel slopes for all tested scenarios.

used for the cathode catalyst as well as the membrane. The second scenario studies direct recycling of the ionomer-free PTE after AST, which allows the cost reduction of not only the Ir catalysts but also Ti PTLs, if electrolyzer performance can be regained after simply recoating the degraded PTE. The last scenario is an extension of scenario 2, which investigates performance improvement when a fresh Ir catalyst layer is coated onto the less degraded interface between PTL and flow field.

The polarization curves measured for three different scenarios are shown in Fig. 6, in comparison to the pristine PTE and PTE after 50k AST cycles. Direct comparison is shown in Supplementary Fig. 15. All three recycling scenarios demonstrated recovery of electrolysis performance compared to the post AST PTE and exhibited similar performance to the pristine ionomer-free Ir PTE (Fig. 6a–c). Besides, the measured values of Tafel slopes compared to PTE after AST indicates a recovery of electrode kinetics for all three scenarios (Fig. 6d). Scenarios 1 and 3 exhibit better recovery compared to scenario 2, which indicates a more severe degradation at the PTL/CL interfaces after ASTs. Overall, three scenarios demonstrate the feasibility of facile process to recycle various degraded components of the ionomer-free PTE MEA, which is crucial to create a circular hydrogen economy.

In summary, we present a novel ionomer-free porous-transport electrode (PTE) exhibiting high performance and durability for proton-exchange-membrane water electrolyzers. This ionomer-free Ir PTE

outperforms a traditional PTE by 652 mV at 1.8 A cm$^{-2}$ at a similar loading and exhibits excellent durability of only 29 mV difference at 4 A cm$^{-2}$ after 50k cycles of accelerated stress test. From both the membrane-electrode-assembly and microelectrode measurements, there is a potential ionomer poisoning effect on Ir impacting oxygen-evolution reaction. Eliminating the ionomer from the catalyst layer not only improves performance of the electrolyzer by mitigating the effect of ionomer poisoning, but also significantly eases processes of fabrication and recycling. The design of the ionomer-free Ir PTE enables facile replacement of electrodes after long-term operation, with performance recovery to that of the pristine electrolyzer. This ionomer-free Ir PTE offers a promising paradigm shift in the electrolyzer industry that potentially enables successful deployment of GW-scale plants to provide cost-efficient clean hydrogen for decarbonization in various sectors.

## Methods

### Ir PTE fabrication

The ionomer-free iridium porous transport electrode is prepared by sputtering Ir onto a commercially available sintered titanium powder-based PTL (Mott Corp.). Prior to Ir sputtering, the PTL was cleaned using a commercial etchant (Multi-etch) for 2 minutes, rinsed in Milli-Q deionized water (18.2 MΩ·cm) for 2 minutes, and was left to air dry. Ir was sputtered onto the PTL using AJA Radio Frequency Sputtering

System (AJA International Inc.) equipped with an Ir target (99.999%, Kurt J. Lesker). The deposition rate of 1.75 Å sec$^{-1}$ was obtained at 3 mTorr under Argon atmosphere. Ir coating duration was controlled for 3, 5, 10, and 20 min to achieve targeted loadings of 0.033, 0.05, 0.085, 0.187 mg$_{Ir}$ cm$^{-2}$, respectively. The iridium loading was measured using X-ray fluorescence spectroscopy (Bruker M4 Tornado, Bruker).

The laser ablated ionomer-free Ir PTE was fabricated by applying Ir coating to a laser-ablated commercial PTL. A class 4 fiber laser cutter (FIBER50FC, Full Spectrum Laser) was used for the laser-ablation process. The PTL was ablated in a cross pattern with spacing between each path set to 0.003 in. First 10 passes were applied at power of 25 W at 80 kHz to melt away the titanium phase, and 60 passes were applied at power of 5 W at 80 kHz to remove burrs created from laser ablation and smooth the surface. After the laser-ablation process, the PTL was rinsed first with isopropanol and then with DI water. The PTL was submerged in isopropanol for 30 min and then underwent Ir coating process as described above.

## Traditional PTE and CCM fabrication

Ultrasonic spray coating was used for fabrication of traditional PTEs and CCMs used for this study. The dilute anode ink was prepared by mixing commercially available iridium oxide catalysts with Milli-Q DI water, ethanol, and n-propanol at a ratio of 1:1:2 by volume and adding in perfluorosulfonic acid (PFSA) ionomer solution (Nafion 5 wt%, Ion Power D521). The ionomer to iridium ratio was set to be 0.116 for all traditional anode catalysts used in this study. The catalyst ink was sonicated using a horn sonicator (CEX500, Cole-Parmer) at power of 38% for 30 min under an ice bath. The Sono-Tek ultrasonic spray coater was used for deposition of the catalyst ink; sonicating nozzle was set to 120 kHz. For PTE fabrication, the commercially available sintered titanium powder-based PTL (Mott Corp.) was held under vacuum plate at 80 °C. For CCM fabrication, a 178 μm dry thickness PFSA membrane (Nafion N117, Ion Power) was used in the vacuum plate at 80 °C. Prior to CCM fabrication, the membrane was first soaked in Milli-Q DI water at boiling temperature (100 °C) for 1 h and was soaked in 0.5 M HNO₃ (ACS Reagent, Sigma-Aldrich) for 1 h to remove impurities and to protonate the sulfonic-acid groups. The membranes were rinsed three times to remove the excess acid and were stored in DI water until the catalyst coatings were applied.

The cathode ink was ultrasonic spray coated onto the membrane to fabricate a half-CCM. The cathode catalyst ink was prepared by mixing platinum supported by carbon (TEC10V50E 46.8% Pt, Tanaka) with Milli-Q DI water and n-propanol at a ratio of 1:1 by volume, and in PFSA ionomer solution (Nafion 5 wt%, Ion Power D521). Ionomer to catalyst ratio was set at 0.45 for all cathode catalysts used in this study. The cathode catalyst ink was bath sonicated for 30 min at 10 °C. The Sono-Tek spray coater at 120 kHz sonication rate was used to deposit cathode catalyst onto the membrane. The platinum loading on the cathode side was measured to be 0.1 mg$_{Pt}$ cm$^{-2}$. The Ir and Pt loadings were measured using an XRF (Bruker). The exact loadings were calculated based on a calibration curve measured from six Ir and five Pt standard loadings purchased (Micromatter Technologies Inc.) along with a blank standard (0 mg cm$^{-2}$). Calibration curves are shown in Supplementary Fig. 16. 16 points were measured uniformly across the samples, and the standard deviation was below 6%.

## Electrolyzer cell assembly

A single electrolyzer cell hardware (Fuel Cell Technology, FCT) with a platinum coated single parallel channel flow field on the anode, and a single serpentine channel graphite flow field on the cathode was used for all the experiments. Sintered titanium powder-based porous sheets (Mott Corp.) were used as either anode PTLs or substrates for PTEs. On the cathode side, carbon paper without an MPL (Toray 120) with 5% PTFE was used as the GDL. In the case of PTE configuration, half-CCMs were used for the electrolyzer cell assembly. 20% compression of the cathode GDL was achieved by controlling the ethylene

tetrafluoroethylene (ETFE) gaskets. Electrolyzer cells were torqued uniformly up to 4.5 Nm. The active areas were designed to be 5 cm$^2$ for both PTE and CCM configurations.

## Electrochemical performance testing

A potentiostat (VSP 300, Biologic) equipped with a 20 A booster was used for electrochemical analyses. In-house modified FCT test station was used for conducting electrolyzer testing. Milli-Q DI water was fed into the anode at 80 °C while cathode inlet was capped and produced hydrogen was vented through cathode outlet. The anode water was recirculated for the duration of the experiment. A rod heater was used to maintain the electrolyzer cell at 80 °C. Following sequences were conducted as a break-in process: (1) 10 cyclic-voltammetry (CV) cycles at a scan rate of 50 mV s$^{-1}$ between 1.2 to 2 V. (2) 5 cycles of CV each at a scan rate of 25, 50, 75, 100, and 125 mV s$^{-1}$ between 0.05 to 1.2 V, respectively. (3) 20 cycles of CV at a scan rate of 50 mV s$^{-1}$ between 1.2 to 2 V. (4) Two repeats of galvanostatic polarization curve measured by holding at various currents from OCV to 4 A cm$^{-2}$ with 130 s holds. Once the break-in procedure has been completed, another set of polarization curve was measured from OCV to 4 A cm$^{-2}$ with 130 s holds followed by measurement of galvanostatic EIS at each current step measured from polarization curves between 1 MHz and 100 mHz. The amplitude of the AC current was optimized for each step to ensure a sufficient signal to noise ratio while maintaining a linear system response. After the EIS measurements, 5 cycles of CV each at a scan rate of 25, 50, 75, 100, and 125 mV s$^{-1}$ between 0.05 to 1.2 V were measured, respectively. All electrochemical experiments were conducted at 80 °C.

## Applied-voltage breakdown

An electrolyzer cell potential, $E_{cell}$, consists of the following elements: reversible cell potential, $E_{rev}^0$, ohmic overpotential, $\eta_{ohmic}$, kinetic overpotential, $\eta_{kin}$, and mass-transport overpotential, $\eta_{mt}$:

$$E_{cell} = E_{rev}^0 + \eta_{ohmic} + \eta_{kin} + \eta_{mt} \qquad (1)$$

As oxygen evolution reaction is the sluggish reaction of the two, the contribution of hydrogen evolution reaction to overpotential is neglected[53], kinetics and mass transport of the anode reaction was only considered for this work along with ohmic losses of the whole cell. The reversible cell potential was defined as[54]

$$E_{rev}^0 = 1.2291 - 0.0008456 \cdot (T - 298.15) \qquad (2)$$

where $T$ is cell temperature [K]. Ohmic overpotential was calculated from high-frequency-resistance (HFR) measurements using EIS:

$$\eta_{ohmic} = i \cdot HFR \qquad (3)$$

where $i$ is the applied current density [A cm$^{-2}$] and $HFR$ is high frequency resistance [Ω cm$^{-2}$] measured by fitting $x$ intercept from the Nyquist plot. Kinetic overpotential was calculated by approximating Tafel region governed by oxygen-evolution reaction,

$$\eta_{kin} = b \cdot \log\left(\frac{i}{i_0}\right) \qquad (4)$$

where $b$ is the measured Tafel slope [V dec$^{-1}$] and $i_O$ is the apparent exchange-current density. With these parameters defined, mass-transport overpotential was calculated by subtracting reversible cell potential and overpotentials from the measured electrolyzer cell potential.

## Accelerated-stress-test (AST) protocol

The durability of the ionomer-free Ir PTE was investigated by analyzing electrolyzer performance after series of potential cycles. AST parameters

used in this study has been selected to achieve harsh conditions based on the previous low catalyst loaded AST study conducted by Alia et al. [51]. 5 s hold at two different potentials ($V_{Low}$ = 1.45 and $V_{High}$ = 2.2) were applied to the electrolyzer cell in the form of a square wave, and the total number of cycles was set to 50k. The anode catalyst loading of the ionomer-free Ir PTE was at 0.085 $mg_{Ir}$ $cm^{-2}$. Cyclic voltammetry, polarization curve, Tafel slope, and electrochemical impedances were measured after each 5k cycles to monitor the degradation process. At the end of the testing, ex-situ characterization was done on the ionomer-free Ir PTE, including measurement of remaining catalyst loading via XRF, surface morphology via SEM, and surface catalyst analysis via XRD and XPS.

### Microelectrode study

Microelectrode experiments were conducted using a previously developed customized cell architecture[42,43]. The microelectrode system is a three-electrode system with an iridium microelectrode (75 μm diameter) as the working electrode (Metrohm Ir.75), the counter electrode is a home-made spray-coated platinum GDE with a loading of 0.5 $mg_{Pt}$ $cm^{-2}$ (Ion Power Inc. 60% Platinum on Vulcan – Carbon Paper Electrode), and the reference electrode is also the GDE with 2% $H_2$/Ar at 50%RH, making it an RHE. A schematic of the setup is shown in Supplementary Fig. 4. The membrane (Nafion 211) is the electrolyte connecting the electrodes. The oxygen evolution reaction was measured with 4% $O_2/N_2$ fed to the working and counter electrodes. The temperature of the cell was controlled and set to 30 °C. The experiment was first conducted without an ionomer coating on the microelectrode, with the microelectrode pressed into the membrane with a force of 200 PSI. The experiment was repeated with a coating of 5 wt% Nafion dispersion drop casted to the microelectrode.

### X-ray computed tomography

Surface morphology of the ionomer-free Ir PTE was analyzed via synchrotron X-ray tomography (XCT). Ex situ XCT was conducted at the Advanced Light Source at Lawrence Berkeley National Laboratory (Beamline 8.3.2) with 100% whitebeam with peak energy greater than 50 keV. optical lenses. Total of 1969 projections were obtained over a rotation of 180°. The exposure time achieved was 200 ms. Dark field and flat field images were used to normalize noise in the incident illumination. TomoPy was used to perform 3D reconstructions and were segmented using an in-house developed python code based on Otsu's thresholding.

### Scanning electron microscopy

The coating morphology of Ir on PTL was characterized using scanning electron microscopy (FEI Quanta FEG 250). Freshly fabricated ionomer-free Ir PTEs were placed into the specimen chamber under high vacuum conditions ($<2 \times 10^{-5}$ Torr). The energy level of the beam was set to 10 kV.

### X-ray diffraction and X-ray photoelectron spectroscopy measurements

The chemical composition of the ionomer-free Ir PTE surface was analyzed using a Rigaku Smartlab X-ray diffractometer equipped with a HyPix-3000 high energy resolution 2D multidimensional semiconductor detector. The parallel beam XRD measurements were performed by setting the same Rigaku SmartLab diffractometer to parallel beam mode.

The ionomer-free Ir PTE surface composition was analyzed using XPS Kratos Axis Ultra DLD system. A monochromatic Al Kα source (hν = 1486.6 eV) was used to excite the samples and detailed spectra of the Ir 4 f, C 1 s, and O 1 s core levels were collected. The measurement was performed under ultrahigh vacuum condition ($7.5 \times 10^{-9}$ Torr). Spectral positions were verified using the adventitious C 1 s signal and found to be within 0.3 eV of the expected value. Spectral analysis was performed using CasaXPS analysis software. Note the Ir XPS

measurements after AST was conducted to the small portion of Ir transferred to the membrane side as it would better represent the oxidation state change after AST.

### Focused-ion beam SEM (FIB-SEM)

FEI Versa 3D dual beam FIB-SEM located at Oak Ridge National Laboratory's Low Activation Materials Development and Analysis (LAMDA) laboratory was used for the FIB-SEM. This technique consists of a process of alternating serial sectioning removal of material followed by imaging of the new sample surface with the electron beam. The ion beam is used to erode the surfaces in the z-direction and the electron beam to image the exposed surfaces in the x-y plane.

## Data availability

The data that support the findings of this study are included in the published article (and its Supplementary Information) or available from the corresponding author upon request.

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

## Acknowledgements

The authors acknowledge the Department of Energy–Office of Energy Efficiency and Renewable Energy–Hydrogen and Fuel Cell Technologies Office (DOE-EERE-FCTO) and the H2 from the Next-generation of Electrolyzers of Water (H2NEW) consortium for funding under Contract Number DE-AC02-05CH11231. This research used resources of the Advanced Light Source (ALS), a DOE Office of Science User Facility under contract no. DEAC02-05CH11231. We are grateful to Dr. Dula Parkinson for helping with micro-tomography measurement at Beamline 8.3.2 of ALS. All opinions expressed in this paper are the author's and do not necessarily reflect the policies and views of DOE. N.D., J.K.L., A.Z.W., X.P. has patent Treatment of A Porous Transport Layer for Use in An Electrolyzer pending to The Regents of The University of California (U.S. Patent Application No. 18/332,886).

## Author contributions

J.K.L. performed most of the electrode fabrication, characterization, cell testing and data analysis. A.W.T. and A.M. also helped with electrode fabrication, characterization, and data analysis. G.A. performed microelectrode measurement under the supervision of R.M. and A.Z.W. J.D.A. performed FIB-SEM under the supervision of D.A.C. F.B. performed XPS measurements and data analysis. X.P., N.D. and A.Z.W. conceived the project. All authors contributed to writing. J.K.L. and X.P. were the primary authors of the paper and chiefly responsible for the experimental design and data analysis.

## Competing interests

The authors declare the following financial interests/personal relationships which may be considered as potential competing interests: X.P., A.Z.W., N.D. reports financial support provided by the Hydrogen and Fuel Cell Technologies Office. N.D., J.K.L., A.Z.W., X.P. has patent Treatment of A Porous Transport Layer for Use in An Electrolyzer pending to The Regents of The University of California. The other authors declare no competing interests.
