## [Peer Review File · Nature Communications]

Reviewer comments, first round -

Reviewer #1 (Remarks to the Author):

Proton exchange membrane water electrolyzers are a promising technique for hydrogen production. The high noble metal loadings of the PEMWE hinders the wide commercialization. Many researchers have paid efforts to develop novel catalysts, components for PEMWE. In this study, the authors developed and studied the ionomer-free porous-transport electrode and evaluated Tafel slope with the conventional PTE. The effects of the Ir loading were comprehensively investigated. The structure, performance, interfacial contact, durability, and recyclability were studied and analyzed. The results are very interesting. The novel PTE structure proposed in this work show a promising direction for PEMWE electrode development, and its good durability and recyclability make it very attractive for research in the field. The manuscript has enough content and fits for the journal. While, there are several issues that should be addressed before the consideration. Please see comments below.

1. Line 144, why PTEs can form better contact between PTL and CL? This should be discussed in details.
2. In Fig 2, caption of d is incorrect, which should be the EDS image of c instead of b.
3. Fig S2, it seems there are errors in the caption.
4. Fig. S3, can you provide EDS mapping for the image? Since currently we can only see the morphology and structure of the particles, but don't know what kind of elements are they.
5. Fig. 3 and 4, have you done any repeats of the tests and how is the reproducibility of the PTE's performance?
6. For the effects of ionomer, what is the thickness of the additional ionomer layer? Does this layer fully covered the Ir PTE surface or just partially covered?
7. For the laser ablation, can you quantify the surface area? Since it is not accurate to say Fig. 4f has larger PTL surface roughness and surface area for electrochemical reaction compared to fig. 4e by seeing.
8. In the section of Recyclability of the Ionomer-free Ir PTEs, can you also provide the voltage breakdown for better comparison of the ohmic and diffusion losses analysis.
9. Fig. 6a, typo in the legend, 'Prestine'. Line 278, there is no Fig. 5e. Line 354, I believe 0.037 is a typo. Please double check throughout the whole manuscript to avoid such issues.
10. What is the Ir loading of the traditional PTL and CCM? I cannot find it in the text.
11. Line 437, what is the value of the apparent exchange-current density how to determine it?

Reviewer #2 (Remarks to the Author):

The manuscript presents a novel physical vapor deposition (PVD) method to fabricate an ionomer-free porous transport electrode (PTE), which exhibits excellent performance and durability at low iridium loadings of only $< 0.1 \text{ mg/cm}^2$. The paper also proposes and demonstrates a new concept for recycling aged cell components. This work represents a significant advancement in the development of electrodes for PEM water electrolysis and has the potential to impact both research and industrial applications. The paper provides clear and convincing evidence for the detrimental effect of ionomer on PTE performance through control experiments and microelectrode measurements. All conclusions are well supported by experimental data. Experimental methods are provided with sufficient details for reproducibility.

However, I have the following concerns that need to be addressed before I can recommend its publication:

1. While the paper demonstrates the good durability of the Ir-PTE using square wave potential cycle experiments, it is important to assess the steady-state operation of the cell over time to test the effectiveness of mass transfer during long-term operation. Therefore, I recommend conducting a durability test using constant current/voltage hold for at least several hundred hours.
2. In spray-coated electrodes, the interfaces between the catalyst layer and Ti-PTL are prone to oxidation during operation. Could the authors provide a discussion or experimental data to clarify if the PTE in this study overcomes this issue?
3. In the context of PTEs, there are several previous publications reporting ionomer-free PTE

studies using different coating techniques such as arc plasma deposition (Masahiro Yasutake et al 2020 J. Electrochem. Soc. 167 124523), magnetron sputtering (Hrbek et al, Journal of Power Sources 2023, 556, 232375), or sputtering (<https://www.businesswire.com/news/home/20221006005600/en/Toshiba>). Given the excellent results of the manuscript, the authors should discuss the advantages of their method compared to these previous publications.

Additionally, there is a redundant sentence on line 73 that should be removed:

"While various types of water electrolyzers exist today, they all operate under similar principles – splitting water into hydrogen and oxygen gases using electricity."

Reviewer #3 (Remarks to the Author):

Manuscript NCOMMS-23-08627-T, by Lee et al. addressed the issue of the high loading of Ir used in the anode electrodes of state-of-the-art PEMWEs. In this work, the authors report the preparation of Ir-coated Porous Transport Layers, with very low Ir-loading, using physical vapor deposition (PVD). In addition to its low Ir-loading, the produced porous transport electrode, PTE, does not contain any ionomers or binders, which, according to the authors, prevents catalyst poisoning.

As the authors point out, the use of high loadings of Ir (ca. 2 mg/cm²) is among the most critical hurdles for the implementation of PEMWEs at the GW scale. Many efforts are devoted to the decreases of Ir usage in PEMWEs, and high-performance electrodes with low Ir loadings ca. 0.4-0.2 mg/cm² have appeared in recent literature. In this sense, the results presented in this manuscript are very interesting since they report even lower Ir contents <0.1 mg/cm², without apparent activity or durability penalties.

However, I wonder whether PVD could be a suitable approach for producing electrolyzers in the GW scale. An Ir coating of 0.187 mgIr/cm² takes 20 min, (by the way is there a typo in the durations reported in page 14?). What is the size of the PTL? What is the thickness of the Ir layer and how homogeneous is it? How long would it take to produce the necessary number of PTEs for a GW electrolyzer by using PVD?

How is the actual loading of Ir on the PTE measured? Was it measured directly from the PTE? Was it measured in different regions of the PTE? Is Ir homogeneously deposited onto the PTE? What is the efficiency of the coating process? Are there any Ir losses during the deposition process?

Figure 2 fails to show nanosized Ir particles, the scale (micrometers) simply doesn't allow to see nanosized particles even if actually formed.

The electrochemical performance shown in Figure 3a is rather high for a low loaded Ir electrode. The authors ascribe, in part, the superior performance of the PTE to the absence of ionomer during the preparation. However, as shown in Figure 3a, very similar polarization curves are obtained (can the authors provide error bars?) for the ionomer-free Ir PTE and for the ionomer Ir PTE cells, so the negative effect of the ionomer is not very clear, at least in the PTE configuration. On the other hand, at high current densities (the region at which mass transport issues may control the overall performance of the cell), the performance of the PTE is below than that reported for standard CCMs with low Ir loadings with ionomers in their formulation, see for instance Bernt et al., J. Electrochem. Soc. 167, 124502 (2020).

The authors suggest that ionomer-induced poisoning is responsible for the lower activity of the ionomer containing PTEs, however, the nature of such poisoning is neither demonstrated nor analyzed.

How was double layer capacitance determined? If from CVs, they should be shown. Have the authors determined the double layers for the ionomer Ir PTE and the Ionomer-free Ir PTE?

As shown in Figure 5d, the intensity of the Ir 4f core-level region for the sample after the AST is significantly smaller than that of the sample before the AST. How does this observation fit with the small loss of Ir from the electrode (ca 10%).

As a curiosity, why was the N1s peak chosen for charge correction in the XPS analysis? Do XPS spectra shown any peaks from the PTL (e.g. Ti 2p or Pt 4f)? If so, what is the surface atomic Ir/Ti ratio?

Energy Storage and Distributed Resources Division

May 3rd, 2023

[redacted]

We would like to thank the reviewers for their time and valuable comments. We modified our manuscript, titled: “Ionomer-free and recyclable porous-transport electrode for high-performing proton-exchange-membrane water electrolysis” (NCOMMS-23-08627-T). We have taken the reviewers’ comments very seriously and spent a significant amount of time strengthening our work. The changes that we have made are indicated by the red text in the manuscript. We have detailed our changes and response to the reviewer comments below.

Reviewer #1 (Remarks to the Author):

Proton exchange membrane water electrolyzers are a promising technique for hydrogen production. The high noble metal loadings of the PEMWE hinders the wide commercialization. Many researchers have paid efforts to develop novel catalysts, components for PEMWE. In this study, the authors developed and studied the ionomer-free porous-transport electrode and evaluated Tafel slope with the conventional PTE. The effects of the Ir loading were comprehensively investigated. The structure, performance, interfacial contact, durability, and recyclability were studied and analyzed. The results are very interesting. The novel PTE structure proposed in this work show a promising direction for PEMWE electrode development, and its good durability and recyclability make it very attractive for research in the field. The manuscript has enough content and fits for the journal. While, there are several issues that should be addressed before the consideration. Please see comments below.

Response: Authors thank the Reviewer for their positive feedback on the submitted manuscript. Authors are confident that the quality of the revised manuscript has been significantly improved based on the Reviewer’s comments. Please find the responses below.

1. Line 144, why PTEs can form better contact between PTL and CL? This should be discussed in details.

Response: Thank you for the comment. The following discussion has been added to the manuscript to fully comply with the Reviewer.

Revised Text (Page 5): “Compared to CCMs, PTEs can have advantages in forming a better contact between catalyst layer and PTL by reducing catalyst layer deformation upon membrane swelling, therefore allowing better catalyst utilization especially at low loading.”

Lawrence Berkeley National Laboratory

One Cyclotron Road / Berkeley, California 94720 / phone 510-486-7045

- In Fig 2, caption of d is incorrect, which should be the EDS image of c instead of b.

Response: Very good catch by the reviewer. The corrections have been made to address the errors in Fig 2.

- Fig S2, it seems there are errors in the caption.

Response: Authors thank the Reviewer for identifying the error. The corrections have been made to address the errors in Fig S2.

- Fig. S3, can you provide EDS mapping for the image? Since currently we can only see the morphology and structure of the particles, but don't know what kind of elements are they.

Response: To fully address Reviewer's comment, the cross-sectional image has been retaken with EDS mapping and EDS spectrum. The Fig. S3 has been revised to include the EDS data.

Supplementary Figure 3: The cross-sectional image of the ultrasonic spray coated porous transport electrode (PTE) and surface SEM image of ionomer coated Ir PTE. a Penetration of catalysts observed from cross sectional image of the spray coated PTE. **b** EDS mapping of iridium shows catalyst particles wasted inside PTL. **c** Surface SEM image and **d** EDS mapping of the fluorine in the ionomer coated Ir PTE after PEMWE operation. **e** EDS spectrum of iridium from **b**. **f** EDS spectrum of fluorine from **d**.

5. Fig. 3 and 4, have you done any repeats of the tests and how is the reproducibility of the PTE's performance?

Response: Authors thank the Reviewer for the question. The repeats of the ionomer-coated Ir PTE, Ir PTEs, and laser ablated Ir PTE are as shown in the modified Figures below. In general, PTEs at ultra-low loading ($\leq 0.05 \text{ mg}_{\text{Ir}}/\text{cm}^2$) show slightly higher variability because of the extremely high sensitivity of performance to loadings. However, PTEs at loading of $\sim 0.1 \text{ mg}_{\text{Ir}}/\text{cm}^2$ show very good reproducibility thanks to the high reproducibility of the PVD process. In this process, the operator just needs to load the PTL sample to a stage and follows the standards of operation, which very much minimizes the user-to-user variability and operational error.

Fig. 3: Electrochemical performance of various PTEs. Comparison of **a** polarization curves and **b** kinetic overpotentials. **c** measured Tafel slopes among the ionomer-free PTE, ionomer-coated PTE, and traditional ultrasonic spray coated PTE. **d** polarization curves of the ionomer-free and ionomer-coated Ir for oxygen evolution reaction in a microelectrode setup.

Fig. 4: The impact of catalyst loading on the Ir PTE and the improved electrolyzer performance with enhanced interface via laser ablation. **a** Polarization curves for Ir PTE at 0.033, 0.050, 0.085, and 0.187 $\text{mg}_{\text{Ir}}/\text{cm}^2$. **b** double-layer capacitance measured at different loadings and the laser ablated Ir PTE. **c** polarization curve. **d** kinetic overpotential measured for the laser ablated Ir PTE at 0.085 $\text{mg}_{\text{Ir}}/\text{cm}^2$. The inset image plots Tafel slopes for the laser ablated and baseline Ir PTE (47.2 mV/dec vs 58.3 mV/dec). **e** SEM image of bare PTL surface. **f** SEM image of laser ablated PTL surface.

- For the effects of ionomer, what is the thickness of the additional ionomer layer? Does this layer fully covered the Ir PTE surface or just partially covered?

Response: Authors thank the Reviewer for the question. Due to the porous nature of PTL and the fact that Ti island on PTL surface is not perfectly flat, we cannot accurately measure the thickness of the additional ionomer layer. But just to give a rough estimation, since the average loading of additional ionomer layer is only about $10 \text{ ug}/\text{cm}^2$ (we maintain an ionomer/Ir ratio of 0.116 for the ionomer-PTE samples at Ir loading of $\sim 0.1 \text{ mg}_{\text{Ir}}/\text{cm}^2$), we think the additional ionomer layer thickness is less than $1 \text{ }\mu\text{m}$ (assuming the Nafion density of $1.58 \text{ g}/\text{cm}^3$ under well-hydrated condition: Anal. Chem. 1996, 68, 21, 3793–3796). Because of the very low ionomer loading and the way of coating ionomer layer is done by spraying an ink without catalyst onto PTE surface, there might be some ink intrusion, leading to ionomer coating uniformity issue. However, we believe the ionomer coating uniformly covers most parts of PTE surface, based on the SEM-EDS obtained from the ionomer coated Ir PTE post to the operation. Even after undergoing cell disassembly after operation, the SEM-EDS was able to see uniform layer of ionomer on most of the PTE surface. The following images were added in the Fig. S3 of the SI.

Supplementary Figure 3: The cross-sectional image of the ultrasonic spray coated porous transport electrode (PTE) and surface SEM image of ionomer coated Ir PTE. a Penetration of catalysts observed from cross sectional image of the spray coated PTE. **b** EDS mapping of iridium shows catalyst particles wasted inside PTL. **c** Surface SEM image and **d** EDS mapping of the fluorine in the ionomer coated Ir PTE after PEMWE operation. **e** EDS spectrum of iridium from **b**. **f** EDS spectrum of fluorine from **d**.

7. For the laser ablation, can you quantify the surface area? Since it is not accurate to say Fig. 4f has larger PTL surface roughness and surface area for electrochemical reaction compared to fig. 4e by seeing.

Response: Thank you for the comment. Quantifying the interfacial contact area of the laser ablated PTL is a rather challenging task. Internally, we have tried techniques such as atomic force microscope to measure the enhanced contact area. It was rather unsuccessful due to the large island size of the sintered PTL. We are also evaluating statistical methods such as machine learning algorithms that can differentiate the solid domain and pore domain based on the contrast difference between these domains in SEM images like Fig 4f. This method is far from reliable before we can release it to the public as it requires a lot of training data.

In the current manuscript, we rely on the double layer capacitance to qualitatively demonstrate the enhanced surface area. It is probably more relevant as it represents the electrochemical active surface area in MEA condition. Besides, in the previous work reported by our group (<https://doi.org/10.1016/j.apenergy.2023.120853>), the laser ablation helps enhance interfacial contact between PTL/catalyst-layer, therefore improving performance for PEMWEs in CCM configuration.

- In the section of Recyclability of the Ionomer-free Ir PTEs, can you also provide the voltage breakdown for better comparison of the ohmic and diffusion losses analysis.

Response: To fully address Reviewer's comment, the Figure S11 in the supporting information now includes voltage breakdowns for each scenario.

Supplementary Figure 11: Direct comparison of electrochemical performance of different recycling scenarios. a Polarization curves measured for varying scenarios. Overpotential breakdowns for b scenario 1. c scenario 2. d scenario 3. Performance rebounds back near pristine with recycled catalysts.

9. Fig. 6a, typo in the legend, 'Prestine'. Line 278, there is no Fig. 5e. Line 354, I believe 0.037 is a typo. Please double check throughout the whole manuscript to avoid such issues.

Response: Authors thank the Reviewer for the revisions. Figure 6, Figure reference to 5e, and the typo in Line 354 has been corrected following Reviewer's comment.

10. What is the Ir loading of the traditional PTL and CCM? I cannot find it in the text.

Response: The anode loading conditions of 0.1 mg_{Ir}/cm² has been added to the manuscript for clarification.

11. Line 437, what is the value of the apparent exchange-current density how to determine it?

Response: Very good catch by the reviewer. The apparent exchange-current density is determined by solving for i_0 in the following equation using the measured data in the Tafel region (linear region between 5 to 80 mA/cm²). This technique is widely used in the literature (Schuler et al., Advanced Energy Materials, 2020):

$$\eta_{kin} = b \cdot \log\left(\frac{i}{i_0}\right)$$

The obtained apparent exchange-current density for the baseline Ir PTE at ~0.1 mg_{Ir}/cm², was about: 2.98*10⁻⁴ mA/cm²

Reviewer #2 (Remarks to the Author):

The manuscript presents a novel physical vapor deposition (PVD) method to fabricate an ionomer-free porous transport electrode (PTE), which exhibits excellent performance and durability at low iridium loadings of only < 0.1 mg/cm². The paper also proposes and demonstrates a new concept for recycling aged cell components. This work represents a significant advancement in the development of electrodes for PEM water electrolysis and has the potential to impact both research and industrial applications. The paper provides clear and convincing evidence for the detrimental effect of ionomer on PTE performance through control experiments and microelectrode measurements. All conclusions are well supported by experimental data. Experimental methods are provided with sufficient details for reproducibility.

However, I have the following concerns that need to be addressed before I can recommend its publication:

Response: Authors thank the Reviewer for their positive feedback on the submitted manuscript. Authors are confident that the quality of the revised manuscript has been significantly improved based on the Reviewer's comments. Please find the responses below.

1. While the paper demonstrates the good durability of the Ir-PTE using square wave potential cycle experiments, it is important to assess the steady-state operation of the cell over time to test the effectiveness of mass transfer during long-term operation. Therefore, I recommend conducting a durability test using constant current/voltage hold for at least several hundred hours.

Response: We understand reviewer's concern for the effectiveness of mass transport under steady-state operations. We therefore conducted a short-term durability study at current density of 1.4 A/cm^2 for about 400h. Due to our current testing setup, fresh DI has to be replenished every day or two, which leads to water tank temperature fluctuation and therefore temporary voltage oscillation. During the durability test, there was insignificant degradation. We therefore believe that the mass transfer is effective and sufficient for thin-film catalyst like the PTE in our case, since mass transfer limitation will lead to acute voltage increase during constant current hold. We added the testing results to the supporting information.

Supplementary Figure 15: Short-term durability at constant current hold of 1.4 A/cm^2 using Ir PTE at anode catalyst loading of $0.085 \text{ mg}_{\text{Ir}}/\text{cm}^2$ and cathode loading of $0.1 \text{ mg}_{\text{Pt}}/\text{cm}^2$. Membrane: Nafion 117, cell temperature: $80 \text{ }^\circ\text{C}$. Average degradation rate is $45 \text{ } \mu\text{V}/\text{hour}$. Fresh water had to be replenished, which led to water tank temperature fluctuation and voltage oscillation.

2. In spray-coated electrodes, the interfaces between the catalyst layer and Ti-PTL are prone to oxidation during operation. Could the authors provide a discussion or experimental data to clarify if the PTE in this study overcomes this issue?

Response: Authors thank the Reviewer for the question. We believe the PTE prepared by PVD or ALD might overcome this issue. One of the reasons that the Ti-PTL are prone to oxidation is that some domains of PTL surface are in direct contact with water and ionomer concurrently in conventional PTL/catalyst-layer interfaces. Once the anode is polarized to high potential, Ti-PTL can get oxygen atoms from water and get oxidized at the interfaces over time. The released protons can migrate through the ionomer to cathode. This explains why PTL at the PTL/catalyst-layer interface is more prone to oxidation than PTL at the bulk or close to the PTL/flow-field interface as there is higher chance for it to be in contact with ionomer in these locations, even though they are at the same potential. And this can't be avoided in a conventional PTL/catalyst-layer interface. However, in the case of ionomer-free PTE prepared by PVD or ALD, there is very little chance for the PTL to touch the water and ionomer/membrane as PVD or ALD create an iridium layer that well covers the surface of PTL. A more detailed study can be found by Chang Liu et al. (ACS Appl. Mater. Interfaces 2021, 13, 16182–16196)

3. In the context of PTEs, there are several previous publications reporting ionomer-free PTE studies using different coating techniques such as arc plasma deposition (Masahiro Yasutake et al 2020 J. Electrochem. Soc. 167 124523), magnetron sputtering (Hrbek et al, Journal of Power Sources 2023, 556, 232375), or sputtering (<https://www.businesswire.com/news/home/20221006005600/en/Toshiba>). Given the excellent results of the manuscript, the authors should discuss the advantages of their method compared to these previous publications.

Response: Authors thank the Reviewer for the references. To fully address Reviewer's comment, the following discussion has been added in the Introduction.

Texts Revised (Page 4):

While advancements have been made towards loading reduction using direct electrode deposition techniques such as works by Yasutake et al.,²² and Hrbek et al.,²³ MEA performance is not yet comparable to the conventional catalyst layers, and recyclability of these anode catalysts has been under investigated.

Additionally, there is a redundant sentence on line 73 that should be removed:

"While various types of water electrolyzers exist today, they all operate under similar principles – splitting water into hydrogen and oxygen gases using electricity."

Response: Based on the Reviewer's suggestion, the redundant sentence in line 73 has been removed.

Reviewer #3 (Remarks to the Author):

Manuscript NCOMMS-23-08627-T, by Lee et al. addressed the issue of the high loading of Ir used in the anode electrodes of state-of-the-art PEMWEs. In this work, the authors report the preparation of Ir-coated Porous Transport Layers, with very low Ir-loading, using physical vapor deposition (PVD). In addition to its low Ir-loading, the produced porous transport electrode, PTE, does not contain any ionomers or binders, which, according to the authors, prevents catalyst poisoning.

As the authors point out, the use of high loadings of Ir (ca. 2 mg/cm²) is among the most critical hurdles for the implementation of PEMWEs at the GW scale. Many efforts are devoted to the decreases of Ir usage in PEMWEs, and high-performance electrodes with low Ir loadings ca. 0.4-0.2 mg/cm² have appeared in recent literature. In this sense, the results presented in this manuscript are very interesting since they report even lower Ir contents <0.1 mg/cm², without apparent activity or durability penalties.

Response: Authors thank the Reviewer for their positive feedback on the submitted manuscript. Authors are confident that the quality of the revised manuscript has been significantly improved based on the Reviewer's comments. Please find the responses below.

However, I wonder whether PVD could be a suitable approach for producing electrolyzers in the GW scale. An Ir coating of 0.187 mgIr/cm² takes 20 min, (by the way is there a typo in the durations reported in page 14?). What is the size of the PTL? What is the thickness of the Ir layer and how homogeneous is it? How long would it take to produce the necessary number of PTEs for a GW electrolyzer by using PVD?

Response: Very good question by the reviewer. We certainly understand reviewer's concern of the scalability of PTE fabrication using PVD (sputtering) to GW scale. In our group, we can fabricate these PTEs using PTLs at 100 cm² maximum. This is lab-scale PVD equipment so we can't use it to fabricate PTEs at industrial scale. However, we certainly believe PTE fabrication using PVD is a scalable method. For instance, companies like Toshiba Corporation have developed home-made PVD equipment to fabricate PTE up to 5 m² (please refer to: <https://www.global.toshiba/ww/technology/corporate/rdc/rd/topics/22/2210-01.html> and https://www.global.toshiba/content/dam/toshiba/migration/corp/techReviewAssets/techreview/2018/03/73_03pdf/a03.pdf) while we were preparing the manuscript for our work. So, it is very possible to fabricate PTE for MW or even GW PEMWEs, although it requires redesigning of PVD equipment. The other benefit of PTE fabricated using PVD is that it enables high-performing and durable PEMWE operation at very low Ir loading (0.1~0.15 mg_{Ir}/cm²), as demonstrated by our work and Toshiba Corporation. And this may address the real bottleneck of scaling up PEMWEs to GW due to the high cost and low global annual supply of Ir.

Due to the challenge of measuring Ir coating layer thickness in the PTE form. We instead coat the Ir to Nafion 117 under the exact same condition as how we coat on PTE at loading of 0.085 mg_{Ir}/cm². The thickness of the Ir layer and its homogeneity are shown from the

newly added figure in SI. The thickness of the Ir layer is about 90 to 100 nm for Ir loading of $0.085 \text{ mg}_{\text{Ir}}/\text{cm}^2$, and it is homogeneous across the coated surface.

Supplementary Figure 16: A cross sectional image of the Ir coating. Ir coating at a loading of $0.085 \text{ mg}_{\text{Ir}}/\text{cm}^2$ was applied to Nafion 117 to measure the Ir coating thickness using SEM. **a** low magnification image of the Ir coating. **b, c, d**, high magnification of the Ir coating. The Ir coating has thicknesses between roughly 90 to 100 nm and is homogeneous across coated surface.

How is the actual loading of Ir on the PTE measured? Was it measured directly from the PTE? Was it measured in different regions of the PTE? Is Ir homogeneously deposited onto the PTE? What is the efficiency of the coating process? Are there any Ir losses during the deposition process?

Response: The loadings were measured from XRF, calibrated with commercial Ir and Pt targets. 16 points were measured for each PTE and CCM, as shown in the new Fig S14, which measured standard deviation smaller than 6%. To fully address Reviewer's questions, the following figure and text has been added to the manuscript. Although we

haven't really attempted to quantify the efficiency of the coating process, we believe the coating efficiency is high (>99%) and these is negligible Ir losses. This is because PCD is, in a sense, like a dry coating process. Compared to the conventional wet coating process (e.g., coating catalyst inks to porous substrate), the dry coating avoids Ir loss such as ink spill and residual ink in containers during ink preparation and Ir loss due to ink intrusion into PTL pores. Besides, as the coating is conducted in a sealed chamber, even if there is Ir loss, it's a temporary loss as it can be recollected.

Added Text (Page 15):

The platinum loading on the cathode side was measured to be 0.1 mg_{Pt}/cm². The Ir and Pt loadings were measured using an XRF (Bruker). The exact loadings were calculated based on a calibration curve measured from six Ir and five Pt standard loadings purchased (Micromatter Technologies Inc.) along with a blank standard (0 mg/cm²). Calibration curves are shown in Fig. S12. 16 points were measured uniformly across the samples, and the standard deviation was below 6%.

Added Figure in SI:

Supplementary Figure 12: X-ray fluorescence calibration curves and a sample data set for measuring Ir and Pt loadings. The XRF used to measure anode Ir and cathode Pt were calibrated using measurements from Ir and Pt standards as well as with a 0 mg/cm² blank. **a** Calibration curves obtained

using standards. **b** example of Ir PTE being measured with an XRF. **c** example of XRF measurement data, where measured Ir loading was $0.052 \text{ mg}_{\text{Ir}}/\text{cm}^2$ with standard deviation of 6%.

Figure 2 fails to show nanosized Ir particles, the scale (micrometers) simply doesn't allow to see nanosized particles even if actually formed.

Response: Very good catch by the reviewer. We deleted 'nano' and modified it to 'Microscopically, a thin layer of Ir particles is coated on top of the PTL surface'.

The electrochemical performance shown in Figure 3a is rather high for a low loaded Ir electrode. The authors ascribe, in part, the superior performance of the PTE to the absence of ionomer during the preparation. However, as shown in Figure 3a, very similar polarization curves are obtained (can the authors provide error bars?) for the ionomer-free Ir PTE and for the ionomer Ir PTE cells, so the negative effect of the ionomer is not very clear, at least in the PTE configuration.

Response: Very good catch by the reviewer. We added error bars to Figure 3a.

Fig. 3: Electrochemical performance of various PTEs. Comparison of **a** polarization curves and **b** kinetic overpotentials. **c** measured Tafel slopes among the ionomer-free PTE, ionomer-coated PTE, and traditional ultrasonic spray coated PTE. **d** polarization curves of the ionomer-free and ionomer-coated Ir for oxygen-evolution reaction in a microelectrode setup.

On the other hand, at high current densities (the region at which mass transport issues may control the overall performance of the cell), the performance of the PTE is below than that reported for standard CCMs with low Ir loadings with ionomers in their formulation, see for instance Bernt et al., J. Electrochem. Soc. 167, 124502 (2020).

Response: Very good point by the reviewer. We certainly agree with the reviewer that the performance of baseline PTE can't outperform the standard CCMs at this stage, so we are still working on ways to further improve the PTE performance. One strategy is to enhance the surface roughness of the PTL using laser ablation. In the manuscript, there is about 50~60 mV of voltage reduction at 4A/cm² for both the sintered PTL (Figure 4c) and fiber PTL (Figure S8). Comparatively, the laser ablated PTE performs similarly to the standard CCMs at much lower Ir and Pt loadings. (PTE: anode loading ~0.1 mg_{Ir}/cm², cathode loading ~0.1 mg_{Pt}/cm²; CCM [Bernt et al., JES 2020: iridium black (0.9 ± 0.3 mg_{Ir} cm⁻²) on the anode and of Pt/C (0.3 ± 0.1 mg_{Pt} cm⁻²). It's also worth pointing out the performance of CCM is at cathode pressure of 30 bar while the PTE performance in this manuscript is at ambient pressure.

Fig. R1. Polarization curve showing performance of Ir PTE, Laser ablated Ir PTE, and the CCM (from literature, Bernt et al, JES, 2020).

The authors suggest that ionomer-induced poisoning is responsible for the lower activity of the ionomer containing PTEs, however, the nature of such poisoning is neither demonstrated nor analyzed.

Response: Very good question by the reviewer. We understand that the ionomer-catalyst interaction is a very broad topic and requires a lot more effort to understand these interactions and possible ionomer poisoning effects for iridium catalyst. Given the scope of current manuscript, we limit this part to MEA and microelectrodes studies, which both indicates a *possible* ionomer poisoning effect for iridium. Internally, we have conducted research on the ionomer thin-film model systems, where we spin-cast Nafion ionomers to various iridium surfaces. We observed the Nafion ionomer structural changes (e.g., phase separation and orientation) using in-situ grazing incidence small-angle X-ray scattering and the ionomer water uptake and swelling behavior using thin-film spectroscopic ellipsometer. These studies indicate that the catalyst-ionomer interactions are strongly dependent on the iridium surface chemistry. We also correlated this model system to more realistic systems such as catalyst inks and MEAs polarization behavior. This study is currently close to being finalized, which might be come across for reviewing in the near future. In the meantime, we are also conducting studies to understand how ionomers with different properties (e.g., side chain length, equivalent weight or non-fluorinated) can impact the Ir-PTE electrolysis performance. This part of the work is on-going so we hope the reviewer can allow us more time to better explore the nature of possible ionomer-poisoning effect.

How was double layer capacitance determined? If from CVs, they should be shown. Have the authors determined the double layers for the ionomer Ir PTE and the Ionomer-free Ir PTE?

Response: Authors thank the Reviewer for questions. Double layer capacitance was determined from series of CVs at varying scan rates using current at non-faradic region. Data has been added to the Figure S6.

Supplementary Figure 6: Applied voltage breakdowns and cyclic voltammograms measured from Ir PTEs at lower loadings. **a** applied voltage breakdown at various Ir loadings. Loadings impact ohmic and kinetics overpotentials throughout current densities, while mass transport exacerbates at high current densities, near limiting currents. Measured cyclic voltammograms for **b** 0.033 $\text{mg}_{\text{Ir}}/\text{cm}^2$. **c** 0.05 $\text{mg}_{\text{Ir}}/\text{cm}^2$. **d** 0.085 $\text{mg}_{\text{Ir}}/\text{cm}^2$. **e** 0.187 $\text{mg}_{\text{Ir}}/\text{cm}^2$ and **f** laser ablated PTE at 0.085 $\text{mg}_{\text{Ir}}/\text{cm}^2$.

The double layers for ionomer Ir PTE and the ionomer-free Ir PTE are as shown below. The ionomer coated PTE shows higher capacitance compared to the pristine PTE by a very small margin. This can be due to an extended interface between the PTE and the PEM. However, this extended interfacial contact doesn't translate into improved performance as shown by the MEA and micro-electrode testing, which could further support a possible ionomer poisoning effect.

Fig. R2. CVs and their according double layer capacitances for Ir PTE and ionomer-free Ir PTE.

As shown in Figure 5d, the intensity of the Ir 4f core-level region for the sample after the AST is significantly smaller than that of the sample before the AST. How does this observation fit with the small loss of Ir from the electrode (ca 10%).

Response: Very good catch by the reviewer. After AST, there was a small portion of the Ir layer transferred from the PTE to the PEM side after the cell disassembly. Because of this, the XPS measurement was conducted to the Ir on the PEM side as we think this side should represent better for the Ir chemical valence change after AST as OER reaction would predominantly occur near the PEM. This is the reason why the intensity of the Ir 4f core-level region looks much smaller compared to the pristine PTE. When we quantified the residual Ir after AST, we counted Ir on both the PEM side and the PTE side using X-ray fluorescence.

We made a few changes as below:

Original (page 10):

XPS of the Ir PTE after the AST shows dominant presence of Ir oxides compared to pristine PTE (Fig. 5d), while no obvious Ir oxides peak is observed through XRD, indicating a growth of amorphous surface oxides thickness, which explains the gradual increase of Tafel slopes mentioned above.

Modified (page 10):

XPS of the Ir after the AST shows dominant presence of Ir oxides compared to pristine PTE (Fig. 5d), while no obvious Ir oxides peak is observed through XRD, indicating a growth of amorphous surface oxides thickness, which explains the gradual increase of Tafel slopes mentioned above.

Original (caption of Fig 5d):

XPS measurements of Ir PTEs before and after 50k cycles of AST.

Modified (caption of Fig 5d):

XPS measurements of Ir before and after 50k cycles of AST.

We also added a statement in the XPS measurements section in method:

Note the Ir XPS measurements after AST was conducted to the small portion of Ir transferred to the membrane side as it would better represent the oxidation state change after AST.

As a curiosity, why was the N1s peak chosen for charge correction in the XPS analysis? Do XPS spectra shown any peaks from the PTL (e.g. Ti 2p or Pt 4f)?. If so, what is the surface atomic Ir/Ti ratio?

Response: Very good catch by the reviewer. We apologize for the confusion caused by a typo. We used the C 1s peak to verify the spectral positions of the measured data and not the N 1s. We revised the method section accordingly. The pristine Ir PTE shows a very weak Pt 4f signal leading to a Pt/Ir ratio below 0.02 and a clear Ti 2p signal leading to a Ti/Ir ratio of around 0.6.

We revised the XPS analysis as below in the method section:

The Ir PTE surface composition was analyzed using XPS Kratos Axis Ultra DLD system at room temperature. A monochromatic Al K α source ($h\nu = 1486.6$ eV) was used to excite the samples and detailed spectra of the Ir 4f, C 1s, and O 1s core levels were collected. The measurement was performed under ultrahigh vacuum condition (7.5×10^{-9} Torr). Spectral positions were verified using the adventitious C 1s signal and found to be within 0.3 eV of the expected value. Spectral analysis was performed using CasaXPS analysis software.

We thank you for your consideration.

Sincerely,

Dr. Xiong Peng

Research Scientist
Lawrence Berkeley National Laboratory
Energy Storage and Distributed Resources Division
Phone: (510) 486-7045
Email: xiongp@lbl.gov

Reviewer comments, further round -

Reviewer #1 (Remarks to the Author):

This is a revised manuscript. The authors have addressed all the comments from previous reviewers and the manuscript is in good shape now. The PTE structure developed in this work pointing out a promising direction for electrode development for water electrolyzers, which is very significant for the community. I would like to suggest the acceptance in its current version.

Reviewer #2 (Remarks to the Author):

My concerns have been adequately addressed; the manuscript has been improved to sufficient quality. Therefore, I recommend its publication as it is.

Reviewer #3 (Remarks to the Author):

The authors have addressed all my remarks. The manuscript can be accepted for publication.